# Effects of Benzalkonium Chloride Contents on Structures, Properties, and Ultrafiltration Performances of Chitosan-Based Nanocomposite Membranes

**DOI:** 10.3390/membranes12030268

**Published:** 2022-02-25

**Authors:** Fitri Khoerunnisa, Mita Nurhayati, Noor Azmi Aulia Annisa, Siti Fatimah, Nisa Nashrah, Hendrawan Hendrawan, Young-Gun Ko, Eng-Poh Ng, Pakorn Opaprakasit

**Affiliations:** 1Department of Chemistry, Indonesia University of Education, Setiabudhi 229, Bandung 40154, Indonesia; mita@upi.edu (M.N.); noornazmi@student.upi.edu (N.A.A.A.); hendrawan@upi.edu (H.H.); 2School of Material Science & Engineering, Yeungnam University, Gyeongsan 38541, Korea; fatimah@ynu.ac.kr (S.F.); nisanashrah@ynu.ac.kr (N.N.); younggun@ynu.ac.kr (Y.-G.K.); 3School of Chemical Sciences, Universiti Sains Malaysia, USM, Penang 11800, Malaysia; epng@usm.my; 4School of Bio-Chemical Engineering and Technology, Sirindhorn International Institute of Technology (SIIT), Thammasat University, Khlong Luang 12121, Thailand

**Keywords:** nanocomposite membranes, chitosan, ultrafiltration, permeability, rejection

## Abstract

The effects of benzalkonium chloride (BKC) contents on the structure, properties, and ultrafiltration performance of chitosan-based nanocomposite membranes containing poly(ethylene glycol) and multi-walled carbon nanotube (chitosan/BKC/PEG/CNT) were examined. The membranes were prepared by a mixing solution method and phase inversion before being characterized with microscopic techniques, tensile tests, thermogravimetric analysis, water contact angle, and porosity measurements. The performance of the nanocomposite membranes in regard to permeability (flux) and permselectivity (rejection) was examined. The results show that the incorporation of BKC produced nanocomposite membranes with smaller pore structures and improved physico-chemical properties, such as an increase in porosity and surface roughness (R_a_ = 45.15 to 145.35 nm and R_q_ = 53.69 to 167.44 nm), an enhancement in the elongation at break from 45 to 109%, and an enhancement in the mechanical strength from 31.2 to 45.8 MPa. In contrast, a decrease in the membrane hydrophilicity (water contact angle increased from 56.3 to 82.8°) and a decrease in the average substructure pore size from 32.64 to 10.08 nm were observed. The membrane rejection performances toward Bovine Serum Albumin (BSA) increased with the BKC composition in both dead-end and cross-flow filtration processes. The chitosan/BKC/PEG/CNT nanocomposite membranes have great potential in wastewater treatments for minimizing biofouling without reducing the water purification performance.

## 1. Introduction

A water crisis is one of the most challenging global problems and affects human survival and economic development. Based on the results of an environmental survey, by 2025, about 1.8 million people will live in regions experiencing a water crisis. Almost 70% of the global population is likely to be affected by water scarcity [1]. This is in line with reports [2] that around 4.0 million people in the world live under periods of water shortage every year for at least one month. Many people are also infected with diseases due to pollution and sanitation problems and from drinking water unsuitable for public consumption. In addition, many companies use groundwater in large quantities in industrial processes, meaning that alternative water sources are needed [3].

Membrane technology for water treatments has been widely developed to overcome the problems of water shortages and water pollution [4,5]. Membrane filtration technology is promising for water treatment and desalination, as it provides a high separation performance and energy efficiency and can be applied over wide pH and temperature ranges [6]. Compared to existing conventional technologies, the major advantage of this technology is its simple operation with no additional chemicals, meaning that its energy consumption is minimal [7]. However, membrane fouling is an unavoidable obstacle and challenge in membrane applications because it can cause a decrease in membrane performance and function [8,9], resulting in a reduction in flux. Therefore, the frequent replacement or washing of the membrane is required. This leads to decreased process efficiency because of the increased operating and maintenance costs [10].

Fouling is generally caused by the deposition and growth of organic, inorganic, colloidal, and microbial foulants [11,12] and the adhesion of microbial cells on the membrane surface [13]. Biofouling commonly originates from organic matter from microbial cells. Biofouling makes up more than 45% of all membrane fouling [14]. The biofilm formation on the membrane surface causes an increase in feed flow pressure and energy consumption and can induce membrane biodegradation [11,13]. A variety of methods have been proposed to overcome biofouling problems, including the biological control of feed water by calculating the total direct cell (TDC) assimilable organic carbon (AOC) [15]; the biofilm formation rate (BFR); and the chemical and physical modification of membrane, such as membrane surface grafting [16] and nanoparticle mixing [17]. Among these, modifications of the membranes by adding antibacterial agents to the membrane matrix (solution mixing) [8,9,18,19] are of interest because these are safer, more economical, and proven to deal with biofouling efficiently.

Benzalkonium chloride (BKC) is commonly used as a biocide and phase transfer agent [20]. The compound shows good antibacterial activity and is safe and friendly to the environment [21,22]. In addition, it is also widely used as a hand sanitizer [21]. The compound is effective against various bacteria, viruses, and fungi, even at low concentrations [20]. Surface-coating membranes with BKC is known to reduce biofilm growth [9,23,24,25].

Chitosan is a bio-based polymer derived from chitin. The material is obtained from various crustacean shells, which have the second largest abundance in the world [26]. Chitosan has a chemical structure that is easily modified, antibacterial activity [27], good permeability, and good hydrophilicity, meaning that it is resistant to fouling. However, chitosan membranes have low stability and mechanical strength. Therefore, it is necessary to modify their manufacturing process [28,29]. The addition of polyethylene glycol (PEG) and carbon nanotubes (CNT) has been shown to significantly increase the permeability and mechanical strength of chitosan membranes [30,31].

In our previous study [18], chitosan/PEG/CNT/BKC nanocomposite membranes were developed using an in situ forming method. BKC was employed as an antibacterial agent. Chemical structures, interactions, and membrane structures were characterized. It was found that the BKC compositions in the membranes significantly determined the variation in the pore structures and the antibacterial activity of the membranes. In this study, the characterization and performance testing of chitosan/PEG/CNT/BKC membranes is carried out in the filtration process using a dead-end and cross-flow system. The effect of the morphology, hydrophilicity, and surface charges of the membranes on their performances is systematically investigated.

## 2. Materials and Methods

### 2.1. Materials

Chitosan (MW ~33 kDa, DD = 87.5%), PEG (MW ~6 kDa), BKC, C_6_H_5_CH_2_N(CH_3_)_2_ (C_8_H_17_)Cl), acetic acid (CH_3_COOH, 1%), sodium hydroxide (NaOH, 99%), and Bovine Serum Albumin (BSA, MW ~66 kDa) were purchased from Merck (Darmstadt, Germany). Multi-walled carbon nanotubes (CNT) were purchased from Wako Chemicals (Osaka, Japan).

### 2.2. Fabrication of Chitosan/PEG/CNT/BKC Nanocomposite Membranes

Chitosan/PEG/CNT/BKC nanocomposite membranes were fabricated via solution mixing and casting methods, as described in our previous work [18]. The mass ratio of chitosan: BKC:PEG:CNT is 8572:10:285:1. The casting solution (25 mL) was poured into a glass mold (ϕ 90 mm). It was then dried for 5 d at 25 °C until the membrane was completely dry. The dried membrane was soaked in 1 M NaOH for 1 h to neutralize the acetic acid. The membrane was then washed with distilled water until neutral and dried by delicately pressing the membrane sandwiched between filter paper sheets, drying under room conditions (25 °C, 70% RH) for 2 d, and further drying at 60 °C for 24 h in a vacuum condition.

### 2.3. Characterization of Nanocomposite Membranes

The morphology of the nanocomposite membranes was characterized by Field-Emission Scanning Electron Microscopy (FE-SEM, Hitachi S-4800, Hitachi, Yamaguchi, Japan). The surface roughness of the membranes was investigated using Atomic Force Microscopy (AFM-Park XE-100, Park System, Suwon, Korea). The average roughness (Ra) and the root mean square average of height deviation (Rq) were calculated to evaluate the membranes’ surface roughness [32]. The tensile strength, elongation, and Young’s modulus of the membranes were evaluated on a tensile tester (Shimadzu EZ-EX-500M, Shimadzu, Tokyo, Japan). The hydrophilicity of the membrane was evaluated by water contact angle (WCA) measurement using the sessile drop method. The porosity of the membrane was determined following the dry–wet weight method, which involved calculating the weight difference (between wet and dry membranes) as a function of the membrane weight [33]. The average pore radius (r_m_) on the membranes was examined based on the filtration rate method and calculated based on the Guerout–Elford–Ferry equation [34,35].

### 2.4. Performances of Nanocomposite Membranes

A permeability test was carried out using a dead-end and cross-flow filtration set, with the membranes’ effective area set at 1.96 × 10^−3^ and 1.66 × 10^−3^ m^2^, respectively. The membrane was compacted with the applied pressure of 2 bar for 1 h until the water flux reached a constant value [36,37]. The rejection test was carried out by employing BSA solutions (100, 200, 300 ppm) at pH 7 as the feed using 2 bar pressure. The permeates and feed solutions from both systems were collected, and concentrations were measured using a spectrophotometer (UV-Vis, Shimadzu-1240, Shimadzu, Tokyo, Japan). The maximum wavelength of the BSA solutions was detected at 720 nm. The water flow velocity over time regarding the membrane water permeability [38] and the membrane rejection corresponding to the number of particles that were removed from the feed water [39] were calculated using the equations reported in our previous work [33,40].

## 3. Results

### 3.1. Characterization of Nanocomposite Membranes

FE-SEM was employed to examine the membrane morphology. The SEM images of chitosan/PEG/CNT/BKC nanocomposite membranes in the cross-section part, as illustrated in Figure 1, depict the asymmetric porous structure of the chitosan matrix with the use of PEG as porogen [18,41]. In addition, the incorporation of BKC induced the formation of an elongated porous structure as a result of the flattened stacking of aromatic rings in BKC and CNT structures [18].

The surface roughness of composite membranes with and without BKC at different compositions is examined by AFM, as shown in Figure 2. The membranes show an increase in the surface roughness with the BKC concentration, as indicated by the R_a_ and R_q_ values. The segregation of BKC clusters during the casting process is the main cause of this roughness increase. Due to their amphiphilic nature, BKC molecules tend to form a cluster structure by stacking their non-polar benzene rings, leading to the extension of their quaternary ammonium groups at the cluster’s surfaces to interact and promote compatibility with the more polar chitosan structure [18,42,43].

Mechanical properties of the composite membranes were examined in tensile mode. The stress–strain curves and mechanical properties of the membranes as a function of BKC compositions were compared (see Appendix A). The BKC contents profoundly influenced the mechanical properties of the membranes. Because of the presence of PEG chains as the softer domains dispersed in chitosan (the stiffer matrix), the tensile strength and Young’s modulus decreased slightly as the BKC concentration increased. However, remarkable improvements in the membrane toughness were observed, as evidenced by substantial improvements in the elongation at break values as the BKC concentration increased, ranging from 45% (M0) to 109% (M50). This is due to the hardening effects induced by the elongated dispersed domains of CNT/BKC/PEG clusters in the stiff chitosan matrix, as reported previously [18,44]. This enhancement in the membrane toughness provides additional benefits for their practical use, especially in ultrafiltration applications.

Thermal gravimetric analysis (TGA) was employed to assess the thermal stability of the composite membranes, as shown in Figure 3a. The initial stage of weight loss from 25 to 177 °C (11–19%) was due to the evaporation of moisture absorbed inside the membranes. From 177 to 357 °C, a significant decomposition stage (up to 40%) was shown, corresponding to the deacetylation and depolymerization of chitosan. Finally, the weight loss above 357 °C was likely associated with residual or crosslinked structures [45,46]. The corresponding DTG curves show that the peak temperature of the second step was significantly unchanged at 290 °C. In contrast, a decrease in the peak temperature was observed in the first and the third stages, from 121 to 103 °C and from 417 to 412 °C, respectively. The results indicate that the decomposition due to the deacetylation and depolymerization of chitosan was unaltered by the addition of BKC. However, the binding interactions between water molecules and chitosan or PEG domains may be interrupted by the competing interactions with the quaternary ammonium of BKC, leading to a decreased evaporation temperature. Additionally, the dispersion of BKC clusters in the chitosan matrix may weaken the interactions of the residual structure, leading to a lower degradation temperature [47] and a reduction in the composition of the final residues.

Since membrane permeability is affected by its hydrophilicity, the water contact angle (WCA) of the composite membranes was examined, as shown in Figure 3b. The addition of BKC resulted in an increase in the WCA values (from 57.0° for M0 to 82.7° for MB400), suggesting a reduction in surface hydrophilicity. No significant difference in WCA values was observed in the nanocomposite membranes with different thicknesses. The reduction in hydrophilicity was likely due to the segregation of the elongated clusters of BKC at the surface of the membranes, as described previously. Similar segregation behavior was also reported in [48]. Even though BKC prefers to create clusters enclosed by PEG chains, an enhancement in its content proportional to PEG causes it to accumulate on membrane surfaces. Furthermore, an increase in the membrane surface roughness, as observed in the SEM images, may also lead to a decrease in the surface hydrophilicity. Nonetheless, because the contact angle values of all composite membranes after the BKC addition were still lower than 90°, they were classified as hydrophilic membranes.

The porosity of nanocomposite membranes was measured. The results, as shown in Figure 4, indicate a slight increase in the value with the BKC concentration. The higher the membrane porosity, the larger the flux obtained was. These enhancements were likely due to the contribution of the polar groups of BKC that accumulate in the polymer membrane structure, causing electrostatic repulsion between the polymer chains. In contrast, the average pore size of the substructure of the nanocomposite membranes declined with the increase in the BKC content, especially at a BKC concentration of 400 ppm. This was likely due to the distribution of the stacking structure of the planar aromatic ring in BKC [18,42], resulting in nanopore blocking. It was noted that the porosity and average pore diameter of the composite membranes were not thickness-dependent (see Appendix A), as the values remained the same for each membrane series at different thicknesses (only 10 μm difference).

### 3.2. Membrane Performance Tests

The membrane performance in ultrafiltration using the dead-end filtration method is summarized in Figure 5. The membrane permeability represented by the deionized water flux was significantly reduced (130 (M0) to 16 L/m^2^ h (M400)) upon the incorporation of BKC and the increase in membrane thickness (130 to 83 L/m^2^ h (0.05 mm); 124 to 59 L/m^2^ h (0.06 mm); 87 to 28 L/m^2^ h (0.07 mm); 16 to 4 L/m^2^ h (0.08 mm)). The permeability value for each membrane series was determined from the constant water flux vs. time (as summarized in Appendix A). The reduction in hydrophilicity and pore size of the composite membranes upon the BKC addition resulted in increased surface tension for the water molecules in their filtration through the pores. In contrast, the tortuosity of the membrane might have been increased due to the reduction in the pore size [49]. Consequently, the water permeability decreased, preventing the passage of water molecules through the membrane [50]. It was also expected that the thicker membranes and those with higher tortuosity had lower flux with a higher rejection ratio during the filtration process. Because water transport into the pore channels is characteristically laminar, flow resistance is caused by the resistance between water molecules and the inner surface of the membrane pores. Therefore, the thicker the membrane is, the greater the mass transport interruption is, but with a lower permeability.

The results obtained for the pressure-dependent water flux from the cross-flow filtration method using membranes with various thicknesses and applied pressures are summarized in Figure 6. The water flux increased as the pressure increased from 88 (2 bar) to 138 L/m^2^ h (4 bar). The results indicate that the incorporation of BKC and increase in the membrane thickness notably reduced the deionized water flux. The morphology of the polymeric membranes was slightly altered due to the pressure driving force, resulting in decreased volume porosity, increased membrane friction, and reduced water permeability. These comprehensively impacted the separation efficiency. Similar to the dead-end method, the decrease in water flux was induced by the increase in the BKC concentration and the membrane thickness. The incorporation of BKC led to a reduction in the membrane hydrophilicity and pore size. The water permeability was reduced as a result of the higher surface tension for water molecules in filtering via the pores, preventing water molecules from penetrating the membrane [51]. In addition, the increase in BKC concentration produced a membrane with a smaller average substructure pore size (Figure 4). The water flux of the membrane was predicted to increase in a similar pattern as the membrane pore radius, following the sieving mechanism theory [52].

During the ultrafiltration process, thicker membranes and smaller pore sizes were also expected to show a lower permeability and a greater rejection ratio [53]. Flow resistance is created by friction between water molecules and the inner surface of membrane pores because water transport into the pore channels is often laminar. As a result, the longer the mass transit delay is and the lower the permeability is, the thicker the membrane will be. It was noted that at the same applied pressure (2 bar), the water flux values obtained from the cross-flow method were underestimated compared to those from the dead-end filtration method.

Similar to the dead-end filtration system, the decrease in the pure water permeability was due to the reduction in hydrophilicity, average substructure pore size, and surface roughness of the membranes with the BKC addition. This caused a decrease in the water penetration ability, as well as the blockage of membrane pores due to the accumulation of hydrophobic planar groups on the membrane surface [18,42]. Since current flows can form hydrodynamic pressure that can control fouling accumulation on the membrane surface, the flux value becomes more stable in cross-flow systems. The permeability value of the composite membrane derived from the cross-flow method was determined by constant water flux vs. time (see Appendix A).

The membrane rejection of BSA was examined using dead-end and cross-flow filtration methods, as summarized in Figure 7 and Figure 8. A similar tendency was observed between the two systems. The membrane rejection performances shown towards BSA increased with the BKC composition. Two factors contribute to this behavior. First, electrostatic interaction between positive charges on the membrane surface and negatively charged BSA molecules (isoelectric point = 4.4) causes the accumulation of BSA on the membrane surface. Second, BSA has a large molecular structure (M_w_ = 66 kDa, hydrodynamic diameter = 6.8 nm [54]), thus hindering the interactions occurring on the membrane surface. The rejection of the composite membrane proportionally increased with the BSA concentration. Increased rejection is associated with a decrease in the average pore size of the membrane and pore-blocking owing to the accumulation of BKC on the membrane surface. The small pore diameter causes the surface area available for BSA adsorption to be larger, thereby increasing the rejection percentage.

The dead-end filtration exhibited slightly lower rejection values compared to the cross-flow filtration. Some solutes may linger behind the membranes, whereas the water penetrates the membrane in dead-end filtration. Accordingly, there is greater friction when passing the membrane. When the pressure of the feedwater is steady, the permeate flux declines. Eventually, the flow significantly decreases and the cleaning of the membrane is required. In dead-end filtration, the membrane pores are blocked and the retentate remains inside. This mode of filtration enables concentrating the solution very rapidly but promotes tough fouling. In cross-flow filtration, the solution is distributed along the membrane with a high shear rate, which restricts the membrane fouling [55]. The feed has no obstacle to its flow, and the retentate can be collected. The separation becomes more intensive and leads to higher rejection than the dead-end method. The results firmly indicate that the nanocomposite membranes have a high capability in wastewater treatments for decreasing biofouling without reducing the water separation performance (see Appendix A).

## 4. Conclusions

The effects of the content of benzalkonium chloride (BKC) on the structures, properties, and ultrafiltration performances of chitosan/PEG/CNT composite membranes were studied. The incorporation of BKC was found to remarkably modify the structure, morphology, physical properties, and performance of the membranes. SEM images showed that the modified membranes had an asymmetric pore structure. In addition, the porosity, surface roughness, and mechanical strength of the membranes increased with the BKC contents. In contrast, the surface hydrophilicity of the membranes decreased with the increase in the BKC compositions, causing a reduction in the membrane permeability. Additionally, the increase in membrane thickness also reduced the membrane water flux. The membrane rejection performances shown toward BSA were improved with the increase in BKC composition in both dead-end and cross-flow filtration due to the decrease in the average pore size of the membranes and pore blocking by the accumulation of BKC clusters. From an environmental viewpoint, this work is highly beneficial, as nanocomposite membranes have great potential in wastewater treatments for minimizing biofouling without reducing the water purification performance.

## Figures and Tables

**Figure 1 membranes-12-00268-f001:**
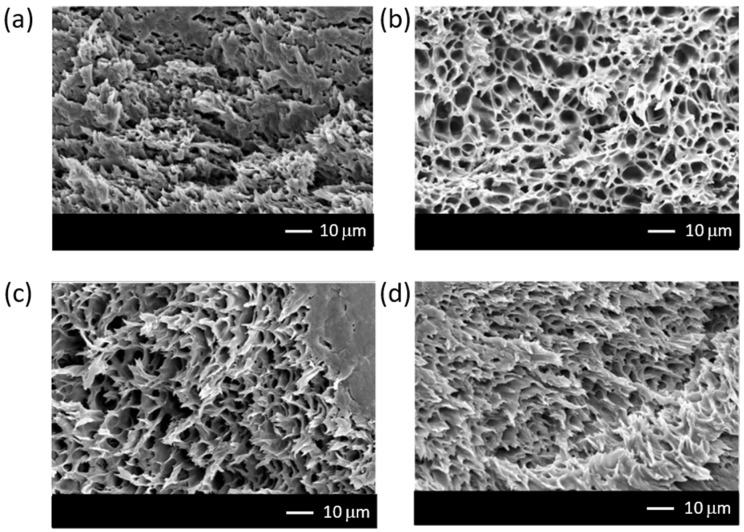
The cross-sectional Scanning Electron Microscopy (SEM) images of chitosan/PEG/CNT/BKC nanocomposite membranes: (**a**) M0, (**b**) M50, (**c**) M100, and (**d**) M400.

**Figure 2 membranes-12-00268-f002:**
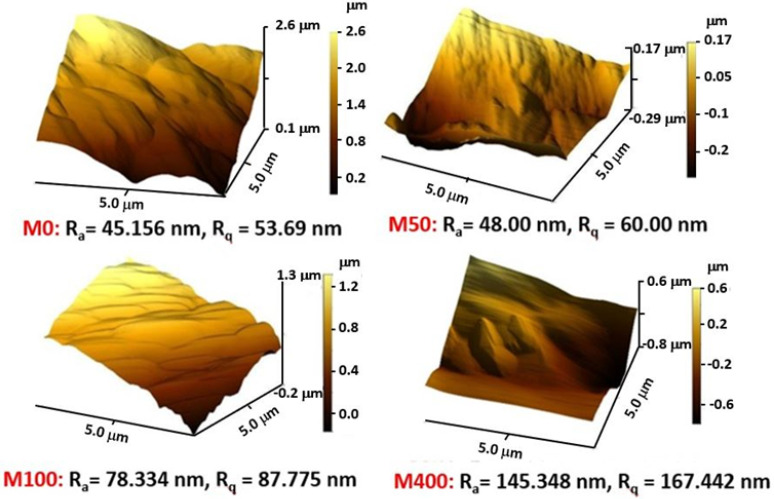
Atomic Force Microscopy (AFM) images of chitosan/PEG/CNT/BKC nanocomposite membrane.

**Figure 3 membranes-12-00268-f003:**
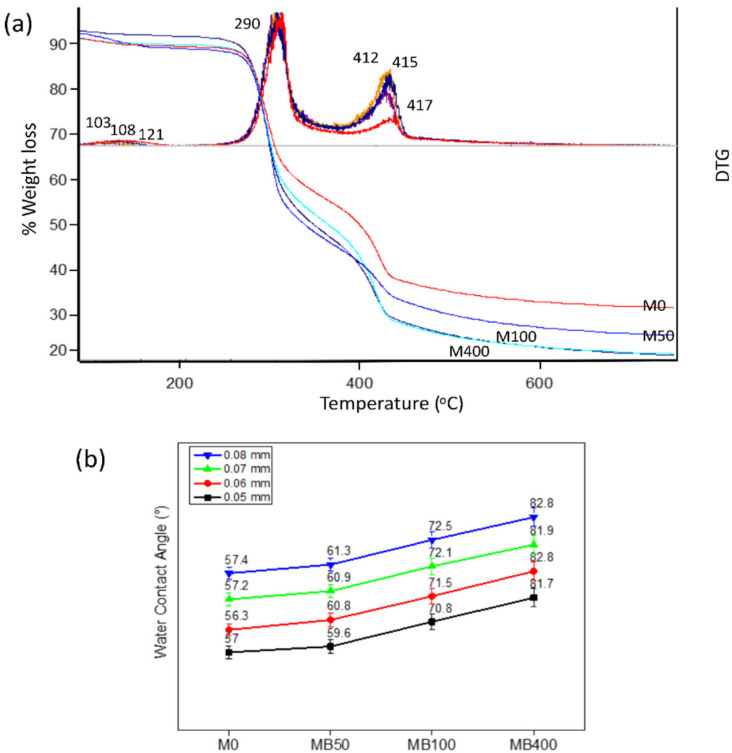
(**a**) Thermogravimetric/differential thermogravimetric (TG/DTG) curves and (**b**) water contact angle values of nanocomposite membranes containing different BKC compositions and different thicknesses.

**Figure 4 membranes-12-00268-f004:**
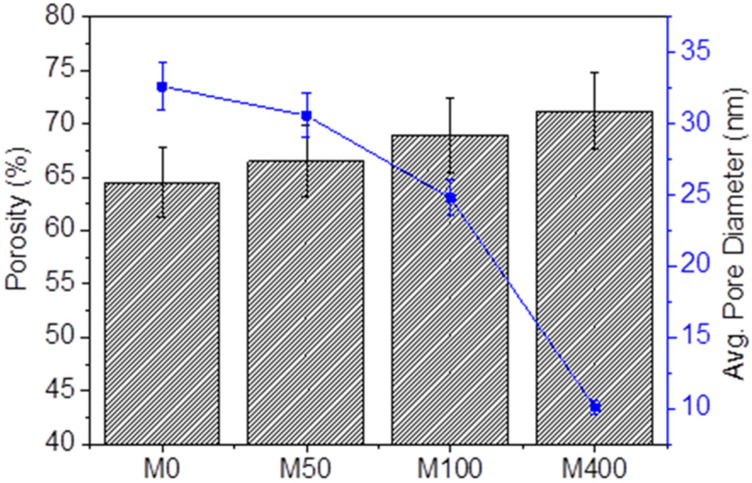
The porosity and average pore size of nanocomposite membranes.

**Figure 5 membranes-12-00268-f005:**
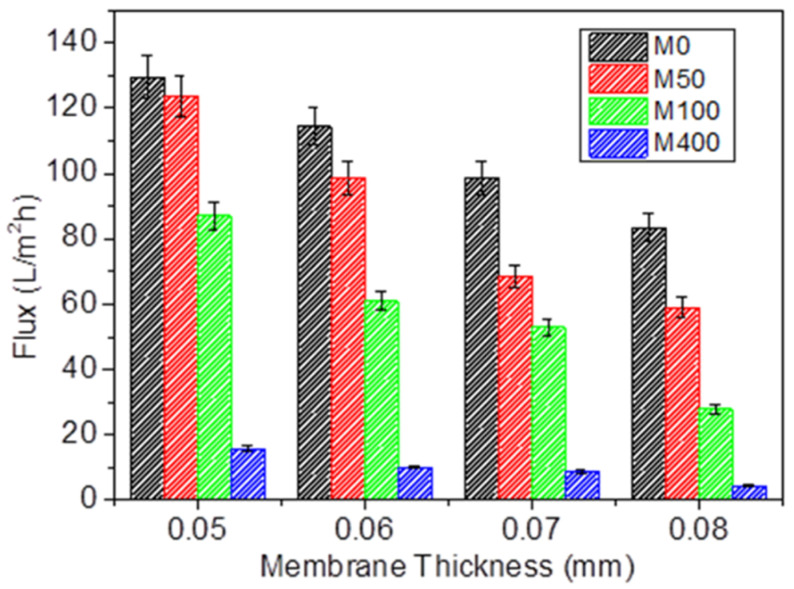
The permeability of nanocomposite membranes with different thicknesses based on the dead-end system at a 2 bar pressure.

**Figure 6 membranes-12-00268-f006:**
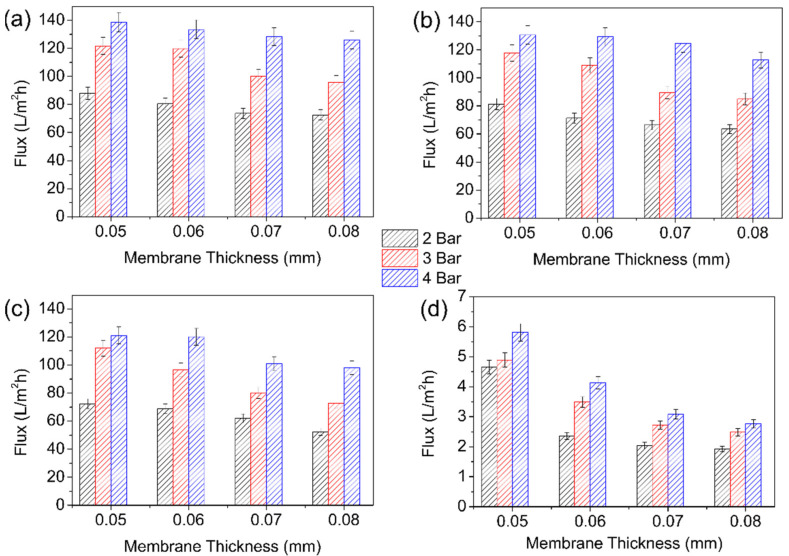
Pressure dependence water flux of nanocomposite membrane using cross-flow method at various membrane thicknesses based on a cross-flow system: (**a**) M0, (**b**) M50, (**c**) M100, and (**d**) M400.

**Figure 7 membranes-12-00268-f007:**
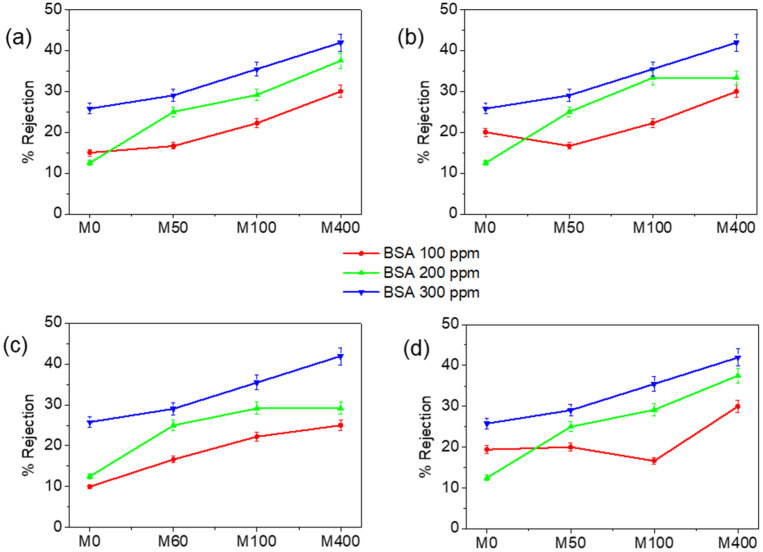
Rejection of nanocomposite membranes of (**a**) 0.05 (**b**) 0.06, (**c**) 0.07, and (**d**) 0.08 mm thickness in a dead-end filtration system at 2 bar pressure.

**Figure 8 membranes-12-00268-f008:**
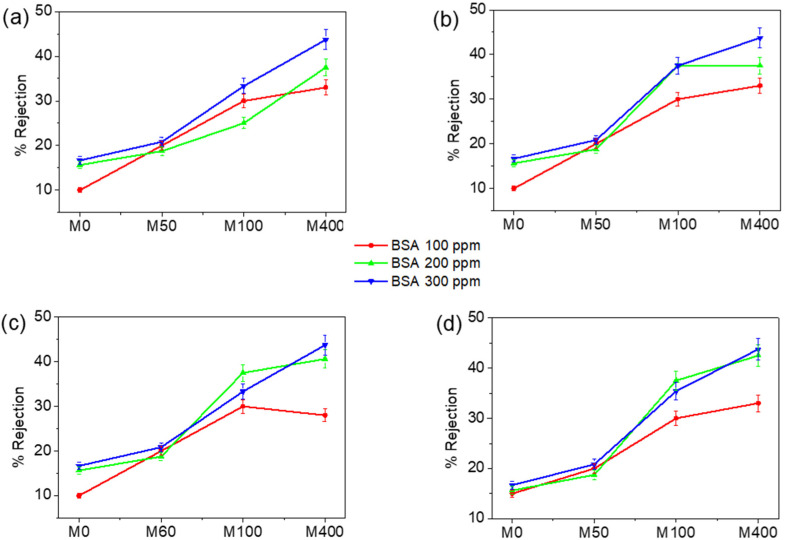
Rejection of nanocomposite membranes of (**a**) 0.05, (**b**) 0.06, (**c**) 0.07, and (**d**) 0.08 mm thickness in a crossflow filtration system at 2 bar pressure.

## Data Availability

The data presented in this study are available on request from the corresponding author.

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
