# Peer review of "Effects of Benzalkonium Chloride Contents on Structures, Properties, and Ultrafiltration Performances of Chitosan-Based Nanocomposite Membranes"

_membranes, 2022, doi:10.3390/membranes12030268_

Round 1

Reviewer 1 Report

In this submitted manuscript, Khoerunnisa et. al. prepared mixed matrix membranes containing chitosan, benzalkonium chloride, MWCNT and PEG and studied the membrane permeability/selectivity properties. Authors evaluate the membranes for water treatment and biofouling efficiency.  Overall, the manuscript is well written and would be a good read for the membrane community. Having said that, I mention below the concerns that are needed to be addresses before the publication of this manscript. I suggest major revision based on following comments.

  1. PEG a porogen added to create porous structure, why is it a part of the membrane name (Chitosan/PEG/CNT/BKC)? Usually, it is not a part of the integral membrane once the membrane is washed with water to remove PEG. PSF membranes are made by adding PEG but is not a part of the name of membrane. I suggest authors consider renaming the membrane.
  2. Authors mentioned the membrane was dried under room conditions (section 2.2, line 106), the humidity level mentioned is 70% RH, this is too high, usual room humidity level is 30-50%. Can authors confirm this? High humidity may affect the nanoscale roughness of the membrane.
  3. Detailed characterization and analysis is provided by the authors, giving a good insight material properties observed.
  4. Authors support the decrease in pore size observation after addition of BKC by stating that the stacking structure of the planar aromatic ring in BKC may block the nanopore and cite reference number 42 for this observation. The reference cited did not use aromatic charged moiety for improving the antifouling or antibiofouling capabilities. Authors should re-evaluate this justification. I think BKC slows down the water-polymer interaction during membrane washing step creating small pores. The stacking of BKC as an explanation for reduced pore size is not convincing. If authors think that indeed stacking is happening, this should be proved using characterization techniques like SAXS or WAXS.
  5. In line 220, authors mention decreases in water flux due to increase in surface tension as supported by the WCA values. Another reason for this may be the increase in tortuosity, given the fact that the pore size has decreased.
  6. “It is also expected that the thicker membranes have lower flux with a higher rejection ratio during the filtration process” line 223, thickness governs the permeability and the selectivity or rejection. The enhanced rejection maybe due to tortuosity or reduced pore size. Similarly re-consider the line 248, thickness doesn’t affect the rejection/selectivity.
  7. In BSA rejection study, M0, M50, and M100 has a pore diameter of more than 25 nm which almost 3 times higher than BSA diameter (7.1 nm), still authors observed more than 10% rejection. Why?

Authors can look at following papers for explanation and justification:

  • https://doi.org/10.1021/acsami.8b20802
  • https://doi.org/10.1021/la052052d
  • https://doi.org/10.1038/nature05532
  • https://doi.org/10.1021/acsami.9b19387
  1. This reviewer would like to see the FRR, Rr and Rir values for all the membrane (M0, M50, 100, and M400). These are considered to be the figure of merits for evaluating the antifouling abilities of the membrane. In line 291, authors declare that the nanocomposite membrane has higher ability for water treatment, the FRR, Rr and Rir values will quantitatively establish authors’ claim.

Authors can read above mentioned paper for such calculation, explanation and justification.

Other comments:

  1. Authors should re-check the manuscript of sentence construction and words used. For example:
    1. In line 234, “with enlargement of applied pressure” should be “with increasing pressure”.
    2. “polymeric membranes was scantily altered”, replace with appropriate word for “scantily” or reconstruct the sentence.
  2. I strongly suggest the manuscript be edited/reviewed by native English speaker before resubmission.

Author Response

Manuscript ID: membranes-157899.

Title: Effects of benzalkonium chloride contents on structures, properties, and ultrafiltration performances of chitosan-based nanocomposite membranes

Dear Prof. Marharyta Vasylieva

Assistant Editor

Membranes,

Thank you very much for your kind message on January 26, 2022, together with the reviewer's comments. We found that the reviewer's comments are very useful for improving the quality of our manuscript and better presenting and clarifying our findings. We have, therefore, fully revised the manuscript accordingly. Our answers to specific comments and the revisions on the manuscript are summarized below.

We hope that the present version is suitable for publication in Membranes. Thank you very much for your kind consideration and support.

Yours Sincerely,

Assoc. Prof. Dr. Pakorn Opaprakasit

Response to Reviewer Comments

>> The authors would like to express sincere gratitude to the reviewers for their valuable comments and suggestions. These are very helpful for improving the quality of our manuscript and better presenting and clarifying our findings. Below are our specific responses to these comments. The revisions have been shown as yellow highlights in the revised manuscript. 

REVIEWER 1

  1. PEG a porogen added to create porous structure, why is it a part of the membrane name (Chitosan/PEG/CNT/BKC)? Usually, it is not a part of the integral membrane once the membrane is washed with water to remove PEG. PSF membranes are made by adding PEG but is not a part of the name of membrane. I suggest authors consider renaming the membrane.

Response: According to our previous work (RSC advances 2021), it was found that PEG plays a role as a porogen and also exists in the composite membranes by co-initiating nanocluster formation with carbon nanotubes (CNT). The nanocluster structures with elongated-spherical shapes are homogeneously dispersed in the chitosan matrix, leading to enhancements in the membrane's mechanical properties. This is evidenced from the characterization results of the membranes, indicating that PEG is stably incorporated in the chitosan matrix even after washing the membrane with water. Therefore, the composite membranes are named as Chitosan/PEG/CNT/BKC.

  1. Authors mentioned the membrane was dried under room conditions (section 2.2, line 106), the humidity level mentioned is 70% RH, this is too high, usual room humidity level is 30-50%. Can authors confirm this? High humidity may affect the nanoscale roughness of the membrane.

Response: The humidity level reported in the procedures is the actual humidity level in the laboratory where the experiments were conducted. Our laboratory is located in an area (Bandung, West Java, Indonesia) that has high precipitation in a long period and high humidity. However, to control the membrane roughness affected by moistures, we further dried the membrane at 60 °C for 24 h in a vacuum condition (line 106). The information has been added to the revised manuscript.

  1. Detailed characterization and analysis are provided by the authors, giving a good insight material properties observed.

Response: Thank you very much for the positive comments and responses.

  1. Authors support the decrease in pore size observation after the addition of BKC by stating that the stacking structure of the planar aromatic ring in BKC may block the nanopore and cite reference number 42 for this observation. The reference cited did not use aromatic charged moiety for improving the antifouling or antibiofouling capabilities. Authors should re-evaluate this justification. I think BKC slows down the water-polymer interaction during membrane washing step creating small pores. The stacking of BKC as an explanation for reduced pore size is not convincing. If authors think that indeed stacking is happening, this should be proved using characterization techniques like SAXS or WAXS.

Response: The reference (42) was cited in this manuscript to confirm the interactions of quaternary ammonium groups with chitosan that cause the modification of the membrane structures. This evidence is relevant to our results, in which the addition of BKC caused the nanopore blocking, resulting in the decrease in pore size. In particular, the ammonium groups of BKC interact with -OH groups of chitosan, while benzene rings induce the aggregation at the membrane surface via p - p stacking, and hence, modifying the membrane structures. The reference (42) is not used to explain the antifouling or antibiofouling capabilities.

The explanations of BKC (and CNT) stacking structures were derived from XRD results, as reported in our previous work (Khoerunnisa F, Kulsum C, Dara F, Nurhayati M, Nashrah N, Fatimah S, Pratiwi A, Hendrawan H, Nasir M, Ko YG, Ng EP, Opaprakasit P, Toughened chitosan-based composite membranes with antibiofouling and antibacterial properties via incorporation of benzalkonium chloride. RSC Advances. 2021;11(27):16814-22). In particular, a new signal was observed at 2q 15.0 o, corresponding to the BKC structure. This signal intensity increased with an increase in the BKC contents. This signal corresponds to a d-spacing of 0.59 Å, which is likely originated from BKC clusters, generated by the p - p  stacking of its aromatic rings or the BKC ring with those of CNT. This further enhances the compatibility of BKC with the chitosan matrix, leading to uniform dispersions of the nanoclusters in the composite membranes and hence a decrease in the membranes' pore sizes. The reference (https://doi.org/10.1039/ D1RA01830B) has been added to the revised manuscript (Lines 207 and 263) to reflect this issue.

  1. In line 220, authors mention decreases in water flux due to increase in surface tension as supported by the WCA values. Another reason for this may be the increase in tortuosity, given the fact that the pore size has decreased.

Response: We agree with the reviewer. More detailed explanations on the possibility of the increase in tortuosity have been added to the revised manuscript (line 222).

  1. "It is also expected that the thicker membranes have lower flux with a higher rejection ratio during the filtration process" line 223, thickness governs the permeability and the selectivity or rejection. The enhanced rejection maybe due to tortuosity or reduced pore size. Similarly re-consider the line 248, thickness doesn't affect the rejection/selectivity.

Response: The membrane permeation flux is proportional to the porosity and inversely proportional to the membrane thickness and tortuosity. We have added the description in the revised manuscript (Lines 225 and 250).

  1. In BSA rejection study, M0, M50, and M100 has a pore diameter of more than 25 nm which almost 3 times higher than BSA diameter (7.1 nm), still authors observed more than 10% rejection. Why?

Authors can look at following papers for explanation and justification:

https://doi.org/10.1021/acsami.8b20802

https://doi.org/10.1021/la052052d

https://doi.org/10.1038/nature05532

https://doi.org/10.1021/acsami.9b19387

Response: According to the rejection results, it was found that the composite membranes have more than 10% rejection toward BSA with the increase in the BKC composition. Two factors contribute to this behavior. First, electrostatic interaction between positive charges on the membrane surface and negatively-charged BSA molecules (isoelectric point = 4.4) causes accumulation of BSA on the membrane surface. Second, BSA has a large molecular structure (Mw = 66 kDa, hydrodynamic diameter = 6.8 nm) that makes it possible to create aggregation, thus hindering the interactions on the membrane surface. It was found that the higher BSA concentration, the larger the rejection (line 233-277). The information has been added in the revised manuscript, along with selected references as suggested.

  1. This reviewer would like to see the FRR, Rr and Rir values for all the membrane (M0, M50, 100, and M400). These are considered to be the figure of merits for evaluating the antifouling abilities of the membrane. In line 291, authors declare that the nanocomposite membrane has higher ability for water treatment, the FRR, Rr and Rir values will quantitatively establish authors' claim. Authors can read above mentioned paper for such calculation, explanation and justification.

Response: We agree with the reviewer. The measurements of FRR, Rr, and Rir for quantitatively evaluating the antifouling abilities of the membranes will be further conducted, and the results will be discussed in detail in a separate publication.

Other comments:

  1. Authors should re-check the manuscript of sentence construction and words used. For example:
    1. In line 234, "with enlargement of applied pressure" should be "with increasing pressure".
    2. "polymeric membranes was scantily altered", replace with appropriate word for "scantily" or reconstruct the sentence.

Response: These have been revised, as suggested. (Line 236 and 239).

  1. I strongly suggest the manuscript be edited/reviewed by native English speaker before resubmission.

Response: The manuscript has been reviewed by a native English speaker.

Reviewer 2 Report

This paper described preparation of Chitosan/PEG/CNT/BKC nanocomposite membranes and its structures, properties. It was found that the addition of benzalkonium chloride (BKC) is helpful to improve its separation performance.Since the reported results were meaningful in this field, I recommend it to be published after minor revision.

1、It is recommended to supplement the FT-IR data of ultrafiltration membrane.

2、Chlorinated benzalkane amine is a good bactericidal material, but it is easily soluble in water. How to ensure that it exists in the membrane and does not lose in the process of sewage treatment.

Author Response

Manuscript ID: membranes-157899.

Title: Effects of benzalkonium chloride contents on structures, properties, and ultrafiltration performances of chitosan-based nanocomposite membranes

Dear Prof. Marharyta Vasylieva

Assistant Editor

Membranes,

Thank you very much for your kind message on January 26, 2022, together with the reviewer's comments. We found that the reviewer's comments are very useful for improving the quality of our manuscript and better presenting and clarifying our findings. We have, therefore, fully revised the manuscript accordingly. Our answers to specific comments and the revisions on the manuscript are summarized below.

We hope that the present version is suitable for publication in Membranes. Thank you very much for your kind consideration and support.

Yours Sincerely,

Assoc. Prof. Dr. Pakorn Opaprakasit

Response to Reviewer Comments

>> The authors would like to express sincere gratitude to the reviewers for their valuable comments and suggestions. These are very helpful for improving the quality of our manuscript and better presenting and clarifying our findings. Below are our specific responses to these comments. The revisions have been shown as yellow highlights in the revised manuscript. 

REVIEWER 2

  • It is recommended to supplement the FT-IR data of ultrafiltration membrane.

Response: This manuscript provides additional and detailed information on the performances of the membranes, whose structures, interactions, and basic physical properties were reported in our previous work (Khoerunnisa F, Kulsum C, Dara F, Nurhayati M, Nashrah N, Fatimah S, Pratiwi A, Hendrawan H, Nasir M, Ko YG, Ng EP, Opaprakasit P. Toughened chitosan-based composite membranes with antibiofouling and antibacterial properties via incorporation of benzalkonium chloride. RSC Advances. 2021;11(27):16814-22, https://doi.org/10.1039/D1RA01830B). In particular, the results from FTIR spectroscopy and other characterization techniques have been discussed. However, the FTIR spectra are available if necessary to be submitted.

  • Chlorinated benzalkane amine is a good bactericidal material, but it is easily soluble in water. How to ensure that it exists in the membrane and does not lose in the process of sewage treatment.

Response: We agree with the reviewer that this is an important issue. In our previous work, the stability of BKC in the composite membranes and the possibility of this compound to be released from the membranes have been examined and discussed (Khoerunnisa F, Kulsum C, Dara F, Nurhayati M, Nashrah N, Fatimah S, Pratiwi A, Hendrawan H, Nasir M, Ko YG, Ng EP, Opaprakasit P. Toughened chitosan-based composite membranes with antibiofouling and antibacterial properties via incorporation of benzalkonium chloride. RSC Advances. 2021;11(27):16814-22, https://doi.org/10.1039/D1RA01830B). Although the results suggest that BKC is immobilized in the membrane matrix with high stability, the concentration of BKC, which may be leached out from the membranes and soluble in the water, was measured by UV-Vis spectroscopy. The results show that only a small amount (<10%) of the released BKC was detected, whose concentration in the water is much lower than its allowable level in the water.

Round 2

Reviewer 1 Report

Although some of the requirements are fulfilled, like FRR values calculation, I feel that others have done satisfactory work to address majority of comments. I recommend acceptance of the manuscript in current format.